

# The CarbonTracker Data Assimilation Shell (CTDAS) v1.0: implementation and global carbon balance 2001–2015

Ingrid T. van der Laan-Luijkx[1,2], Ivar R. van der Velde[3,4,1,5], Emma van der Veen[1], Aki Tsuruta[6], Karolina Stanislawska[7], Arne Babenhauserheide[8], Hui Fang Zhang[9,10], Yu Liu[11], Wei He[5,12], Huilin Chen[5,4], Kenneth A. Masarie[3,*], Maarten C. Krol[1,2,13], and Wouter Peters[1,5]

[1]Wageningen University and Research, Environmental Sciences Group, Wageningen, the Netherlands
[2]Utrecht University, Institute for Marine and Atmospheric Research, Utrecht, the Netherlands
[3]Global Monitoring Division, Earth System Research Laboratory, National Oceanic and Atmospheric Administration (NOAA), Boulder, Colorado, USA
[4]University of Colorado, Cooperative Institute for Research in Environmental Sciences (CIRES), Boulder, Colorado, USA
[5]University of Groningen, Centre for Isotope Research, Groningen, the Netherlands
[6]Climate Research, Finnish Meteorological Institute, Helsinki, Finland
[7]Meteorological Research, Finnish Meteorological Institute, Helsinki, Finland
[8]IMK-ASF, Karlsruhe Institute of Technology (KIT), Karlsruhe, Germany
[9]State Key Laboratory of Resources and Environment Information System, Institute of Geographic Sciences and Natural Resources Research, Chinese Academy of Sciences, Beijing, China
[10]University of Chinese Academy of Sciences, Beijing, China
[11]Institute of Biogeochemistry and Pollutant Dynamics (IBP), ETH Zürich, Zürich, Switzerland
[12]International Institute for Earth System Science, Nanjing University, Nanjing, China
[13]SRON Netherlands Institute for Space Research, Utrecht, the Netherlands
[*]Currently at: SkyData Solutions LLC

*Correspondence to:* Ingrid van der Laan-Luijkx (ingrid.vanderlaan@wur.nl)

**Abstract.** Data assimilation systems are used increasingly to constrain the budgets of reactive and long-lived gases measured in the atmosphere. Each trace gas has its own lifetime, dominant sources and sinks, and observational network (from flask sampling and in situ measurements to space-based remote sensing) and therefore comes with its own optimal configuration of the data assimilation. The CarbonTracker Europe data assimilation system for $CO_2$ estimates global carbon sources and sinks,

5  and updates are released annually and used in carbon cycle studies. CarbonTracker Europe simulations are performed using the new modular implementation of the data assimilation system which is called the CarbonTracker Data Assimilation Shell (CTDAS). Here, we present and document this redesign of the data assimilation code that forms the heart of CarbonTracker, specifically meant to enable easy extension and modification of the data assimilation system. This paper also presents the setup of the latest version of CarbonTracker Europe (CTE2016), including the use of the gridded state vector, and shows the resulting

10  carbon flux estimates. We present the distribution of the carbon sinks over the hemispheres and between the land biosphere and the oceans. We show that with equal fossil fuel emissions, 2015 has a higher atmospheric $CO_2$ growth rate compared to 2014, due to reduced land carbon uptake in later year. The European carbon sink is especially present in the forests, and is reduced during drought years. Finally, we also demonstrate the versatility of CTDAS by presenting an overview of the wide range of applications for which it has been used so far.





# 1  Introduction

The CarbonTracker data assimilation system for $CO_2$ estimates global carbon sources and sinks and was originally developed at the National Oceanic and Atmospheric Administration's (NOAA) Earth System Research Laboratory (ESRL) in the period 2005-2007 (Peters et al., 2005, 2007). After that, development continued in two separate branches: 1) CarbonTracker

(NOAA/ESRL) and 2) CarbonTracker Europe (CTE, Peters et al., 2010), referring to the location of development. This paper describes the developments in the second branch.

The CarbonTracker data assimilation system for $CO_2$ estimates the carbon exchange between the atmosphere, land biosphere and oceans, using atmospheric observations of $CO_2$ mole fractions. A key element of CarbonTracker is the two-way nested TM5 transport model (Krol et al., 2005; Huijnen et al., 2010) which connects the surface fluxes to atmospheric $CO_2$ mole

fractions. The existing code base of TM5 in Fortran was, in 2005, also the basis for CarbonTracker, requiring relatively little additional code to apply it as a $CO_2$ ensemble Kalman smoother, since over 90% of the computational time of a CarbonTracker simulation was spent on the TM5 transport model. Over time though, new requirements for CarbonTracker arose, specifically requiring new and more complex data structures and work flows to be handled, which were cumbersome to implement in Fortran, and not always compatible with the ongoing development of TM5. Many of these new requirements could be easily

accommodated in a more versatile data assimilation framework. This lead to the new object-oriented implementation in the Python programming language and is called the CarbonTracker Data Assimilation Shell (CTDAS). It is designed in a modular fashion that allows for new observation types to be introduced, changes in the structure of the underlying state vector to be made, and even replacement of the transport model (e.g. the Lagrangian model STILT) or the optimization method (e.g. four-dimensional variational (4DVar)), with only minimal additional code within one module. Sect. 2 documents the new code and

its possibilities.

In Sect. 3 we describe the setup of the latest version of CarbonTracker Europe for $CO_2$ (CTE2016) and present its results, including carbon flux estimates that have been used in several carbon cycle studies. CTE2016 is based on the original CarbonTracker, of which one of the shortcomings concerns the relatively coarse set-up of the state vector. This state vector contained scalar multiplication factors for a maximum of 240 "ecoregions": broad distributions of vegetation types across continents that

are assumed to have fully correlated errors over their geographical extent. Although this choice represented a leap forward in 2007, when observations were sparse and most other inversion systems were even coarser, it has now become possible to replace it with a "gridded" state vector. In this approach, each element of the Earth's surface (typically resolved at $1°x1°$) is more or less independent, depending on pre-set correlation length scales and the correlation e.g. decays exponentially with distance. In Sect. 3.2 we will also show the implementation of this gridded state vector with minimal changes to the code and

assess its impact on estimated $CO_2$ surface fluxes.

Since we have already demonstrated the power of the CarbonTracker system in previous work (Peters et al., 2005, 2007, 2010), we focus here on new extensions and applications of CarbonTracker Europe, which also demonstrate the power of CTDAS. We therefore do not include observation system simulation experiments (OSSEs) which are traditionally presented alongside the implementation of a data assimilation system. CTDAS is currently used in at least seven institutes that perform





ensemble data assimilation of trace gases, with applications in $CO_2$, $CH_4$, $^{13}CO_2$, carbonyl sulfide (COS), and $SF_6$. These applications have helped to improve its code base and test its implementation in several setups. We will show an overview of the current applications in Sect. 4.

In this paper we 1) document the CTDAS code base, 2) present the setup of the latest version of the CarbonTracker Europe (CTE2016), together with the resulting carbon flux estimates, and 3) demonstrate the versatility of CTDAS by presenting an overview of the applications it has been used in so far.

## 2 CTDAS design and implementation

### 2.1 Data assimilation in CarbonTracker

The CarbonTracker data assimilation system for $CO_2$ estimates carbon fluxes between the atmosphere and the surface (land biosphere and oceans), using observations of atmospheric $CO_2$ mole fractions. At its core, CarbonTracker is an ensemble Kalman smoother application using a fixed-lag assimilation window (Peters et al., 2005) of which several flavors are used in trace gas studies (e.g. Prinn et al., 1995; Zupanski et al., 2007; Bruhwiler et al., 2005). The surface $CO_2$ fluxes are optimized using the cost function ($J$) that describes the system according to:

$$J(x) = (\mathbf{y^o} - \mathcal{H}(\mathbf{x}))^T \mathbf{R}^{-1}(\mathbf{y^o} - \mathcal{H}(\mathbf{x})) + (\mathbf{x} - \mathbf{x^b})^T \mathbf{P}^{-1}(\mathbf{x} - \mathbf{x^b}) \qquad (1)$$

Where $\mathbf{y}$ are the atmospheric $CO_2$ mole fraction observations, with their covariance $\mathbf{R}$. $\mathcal{H}$ is the observation operator (TM5) that connects the observations $\mathbf{y^o}$ to the scalars that modify the surface $CO_2$ fluxes, which are contained in the state vector $\mathbf{x}$. Prior information on the surface fluxes is contained in the background state vector $\mathbf{x^b}$ with covariance $\mathbf{P}$. Ensemble statistics are created from 150 ensemble members, each with its own background $CO_2$ mole fraction field. We have restricted the length of the smoother window ('lag') to only five weeks as we found that the derived flux patterns within Europe and North America are robustly resolved well within that time. We refer the reader to previous publications (Peters et al., 2005, 2007, 2010) and the webpage (http://www.carbontracker.eu/documentation) for further general details on the ensemble Kalman smoother as applied in CarbonTracker.

### 2.2 Motivation for CTDAS

CarbonTracker started with $CO_2$ data assimilation included in the TM5 Fortran code. With ongoing developments in CarbonTracker, we required a more flexible data assimilation framework, that could accommodate more complex data flows and structures, and be applied to other applications. Such frameworks for data assimilation exist, and have been successfully used across a range of applications. One example of a popular data assimilation package is the Data Assimilation and Research Testbed, DART (see http://www.image.ucar.edu/DAReS/DART, Anderson et al., 2009; Raeder et al., 2012). It offers many out-of-the-box options for data assimilation and supports a wide range of platforms and possible applications. These are primarily, but certainly not limited to, meteorological data assimilation efforts and include ensemble systems oriented on atmospheric constituents (e.g. Arellano et al., 2010). Another example is the openDA toolkit resulting from initial developments at Delft

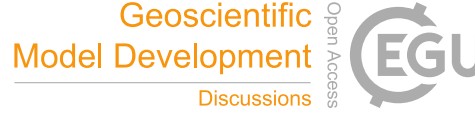



University (http://www.openda.org/joomla/index.php) which initial focus was on hydrological applications, but was expanded to also include wave models and air quality models. Furthermore, ECMWF developed the Object-Oriented Prediction System (OOPS) framework (Tremolet et al., 2013) which is used in their Integrated Forecasting System (IFS). These open source frameworks aim to provide their users with an easy-to-use and well-documented data assimilation system, and in that sense

would be suitable for CarbonTracker as well. However, the CarbonTracker system is characterized by a long lag window of several weeks, and by a very expensive observation operator (i.e. a TM5 simulation). Since the application of an ensemble Kalman smoother is also not provided by any existing open source system, we decided to implement our own data assimilation shell.

Looking at the requirements for our CarbonTracker Data Assimilations Shell (CTDAS), we realized that the Python language

could handle the tasks needed such as basic shell scripting, use of numerical recipes, job control under UNIX, I/O in NetCDF and HDF, analysis and visualization, and even remote interfacing over TCP/IP and HTTP. Pythons' functionality for object-oriented implementation moreover suited well our desired modular design of CTDAS, with minimal code duplication and efficient use of class inheritance to build diverse pipelines for data assimilation. Specifically, we aimed to make CTDAS:

- Independent of application (carbon dioxide, methane, isotope ratios, or multi-tracer);

- Independent of data assimilation design (choice of state vector and observations, or optimization method for cost function minimization);

- Independent of observation operator (e.g. atmospheric transport models like TM5, WRF, STILT, biogeochemical SiB-CASA, or combinations of these);

- Extendible, documented, open-source (GNU GPLv3), multi-platform.

The choice to build a custom data assimilation shell for CarbonTracker and to implement it in Python, led to the development of CTDAS as presented here. The next sections provide more detailed information on the CTDAS code, including the design and implementation.

## 2.3 Modular structure of CTDAS

The CTDAS code is based on the use of seven Python classes[1], each representing a different part of the data assimilation

system. They are visualized in Fig. 1. Three classes are referred to as "control" classes, as the objects they instantiate are used to control the ensemble data assimilation system. These three control classes are:

1. **Class CycleControl**: controls the cycling through time, succession of cycles, and organization of input and output data, including checkpointing data, for each cycle. This is the only core object of CTDAS that is automatically created based on options and arguments passed along when submitting the main CTDAS job (e.g. cycle length, smoother window

length (lag), and number of ensemble members).

---

[1]see e.g. http://en.wikipedia.org/wiki/Class_(computer_programming) for an explanation of the object-oriented programming terminology used in this section.





2. **Class DaSystem**: describes the characteristics of the current data assimilation system in terms of state vector size, covariances and locations of input files.

3. **Class Platform**: controls operations specific to each computing platform such as submitting jobs to the queue, creating directories, and settings of the environment.

The specific details for a given experiment are controlled through external run-control files (rc-files), which consist of key:value pairs that pass information to CTDAS on e.g. the dates for which to run the experiment or the number of parameters (scaling factors) and ensemble members. For each of the three control classes CTDAS provides a "base class" describing the required methods, attributes and the expected interface when accessing these from within CTDAS. Specific applications can then inherit these base classes, and modify only those methods or attributes that differ for their specific configuration. For
example, a Platform object with a method to submit a job script with a proper command (e.g. sbatch) to a specified queueing system (e.g. SLURM) can be used for a high performance computing environment. This same method in the Platform object could similarly prepare a job script for the next cycle on a regular workstation, but in that case could e.g. simply spawn a new task (sh) for this job.

The four classes that complete CTDAS are:

4. **Class StateVector**: builds the data structure of a state vector, defined by 3 dimensions in parameter space (number of scaling factors, ensemble members and lag), including sampling of random ensemble members from a specified distribution.

5. **Class Observations**: reads observational input data and prepares the observations to be used by the observation operator. Observation specific information (e.g. model-data mismatch values) is defined in and passed from an rc-file.

6. **Class ObservationOperator**: controls the sampling of the state vector (e.g. simulating mole fractions), including e.g. the setup, compilation and calling of the transport model.

7. **Class Optimizer**: handles the optimization of the state vector (using e.g. a minimum least squares method) given a set of observations.

These seven classes represent the typical components of a data assimilation system. They are imported as objects in the
main Python script and can take on many different formats depending on the application. Because the information in the Observations and StateVector classes are different for nearly every application, their dimensions and the reading of data are controlled through external rc-files that specify how to construct the corresponding objects. For the Observations class, this could for instance look like:

```
– species: co2
```
```
– input.dir: /myfolder/observations/co2/
```
```
– input.file: $input.dir/obspack_v1.0.nc
```





This external control makes it easier to use settings consistently across experiments, and also precludes the need to hard-code these basic properties for each application. As long as the objects that are instantiated can parse the provided rc-file and properly populate itself with the data, the system will work.

The class Optimizer currently supports two versions of the square root ensemble Kalman smoother originally presented in Whitaker and Hamill, (2002) and Peters et al, (2005), both for an observation serial algorithm and a batch algorithm. In the latter, the Kalman filter equations are solved using matrix expressions of $\mathbf{K}$ (the Kalman gain matrix), $\mathbf{R}$, and $\mathbf{HPH}^T$ rather than scalar or vector values. This can be useful when observation errors are correlated (a non-diagonal matrix $\mathbf{R}$). Other optimization methods (e.g. 4D variational approach) have so far not been implemented in CTDAS, but can be added with relatively little effort by creating a new Optimizer class.

Special attention is focused on the ObservationOperator, which consumes the majority of CPU in CarbonTracker, and was previously TM5 by definition because it was the heart of the code base. Here, we have explicitly made the observation operator external to the CTDAS code and call it from a separate class. This allows TM5 to be replaced by a different transport model in CTDAS, and also enables development and maintenance of the TM5 code separate from CTDAS. In the currently implemented TM5 ObservationOperator class, an external call compiles the TM5 transport model (using Fortran and a set of TM5 specific control scripts), and this precompiled TM5 executable is subsequently called to simulate mole fraction needed in the ensemble Kalman smoother. Control of TM5 is taken over by the CycleControl object, which modifies TM5 specific input data for the current data assimilation cycle (e.g. begin and end time). The Platform object allows TM5 jobs to be run in parallel operation through the queuing system, and once finished returns control to the main Python program (CTDAS itself is currently not parallelized). This job flow is further explained in the next section, but we stress here that all references to TM5 in this paragraph can easily be replaced by that of any other transport model (e.g. WRF, GEOS-Chem, or even Lagrangian transport models like STILT) as long as there is an appropriate ObservationOperator class.

### 2.4 Inverse, forward and analysis pipelines

The seven classes described above are imported as objects in the main Python script, which subsequently calls a "pipeline" script with these objects as arguments. The pipeline takes care of the order in which all steps of an experiment are performed. A key property of the pipeline is that all calls to methods in external modules (i.e., function calls) are generic, rather than specific. This means for instance that to achieve a simulation of the transport model, the generic method (e.g. `run_simulation()`) of an ObservationOperator is called rather than an application specific method (such as `run_tm5_with_co2()`). The pipeline will therefore work for any ObservationOperator class with a properly programmed interface, and can be independent of specific implementations of a transport model.

The objects used in CTDAS can not only be tailored to a specific application, but they can also be combined in different ways, yielding different pipelines. An example is the simple "forward" pipeline, which combines the complementary Observations, StateVector, and ObservationOperator objects with the three control classes. The forward pipeline simulates forward transport (ObservationOperator) of a given tracer as controlled by specified inputs (such as emission scaling factors) in the StateVector, while sampling mole fractions at all times and locations included in the Observations object. This sequence is repeated for all


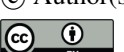

time steps specified in CycleControl, until the final cycle is reached. Another example is the "analysis" pipeline, combining Observations and StateVector objects with the three control classes, to extract the results from an experiment to convenient output formats (e.g. aggregated fluxes for defined regions).

A more complex pipeline, important to this paper, is the inverse pipeline that yields an actual optimization result. The
5  pseudo-code that achieves this in CTDAS (similar to the illustration in Peters et al., (2005)) is:

1. Create the 7 objects from the code structure (note that the first is automatically created from options and arguments when submitting the main CTDAS job, see Sect. 2.3):

```
DaCycle = CycleControl(opts, args)
DaSystem = da.carbondioxide.dasystem()
```
10  
```
PlatForm = da.platform.cartesius()
Observations = da.observations.obspack_obs()
StateVector = da.co2gridded.statevector()
ObsOperator = da.tm5.observationoperator()
Optimizer = da.baseclasses.optimizer()
```

2. Read Observations (x,y,z,t) for this cycle ($y^o$):

```
Observations.read_data(CycleControl.time[0])
```

3. Read or construct StateVector ($x^b$):

```
StateVector.Initialize(CycleControl.time[0])
```

4. Compile ObservationOperator ($\mathcal{H}$):
```
ObsOperator.Compile()
```

5. Run ObservationOperator for nlag cycles, and sample at (x,y,z,t): $\mathcal{H}(x^b)$:

```
for n in range(nlag):
    ObsOperator.Run(CycleControl(time[n])
```

6. Optimize StateVector (from $y^o$, $\mathcal{H}(x^b)$ and Kalman filter equations): $x^a$:
```
Optimizer.serial_least_squares()
```

7. Run ObservationOperator for n=1 and sample at (x,y,z,t): $\mathcal{H}(x^a)$:

```
ObsOperator.Run(CycleControl(time[0])
```

As noted, this pseudo-code uses generic methods of each object and is therefore application independent.

## 2.5 CTDAS documentation and version control

The CTDAS system is documented using the open-source SPHINX package (http://sphinx-doc.org) that can export documentation written inside the code itself to various output formats including HTML, PDF, RTF, and more. The output of CTDAS





documentation can be viewed at http://www.carbontracker.eu/ctdas/. An important advantage of this inline documentation is that the code and its description exist within the same text files, and are thus more easily updated together. This is preferably done at the same time that the source code is modified, by the programmer doing the actual modifications. Because the syntax of this documentation is relatively simple (SPHINX handles the translation to nicely readable document formats), the burden

on code developers is minimal.

## 3   Updates and results from the latest version for CO$_2$: CTE2016

In this section we describe the application of CTDAS for the latest version of the CarbonTracker data assimilation system for CO$_2$: CarbonTracker Europe (CTE2016). We focus on the updates compared to previous versions (Sect. 3.1 and 3.4), specifically related to the state vector (Sect. 3.2). For more general information on CarbonTracker we refer to previous publications

(Peters et al., 2005, 2007, 2010). The differences compared to NOAA's CarbonTracker are included in Sect. 3.1.

### 3.1   General setup for CarbonTracker Europe for CO$_2$

CarbonTracker estimates weekly scaling factors ($\lambda_r$) for both Net Biome Exchange (NBE) and net ocean exchange, using atmospheric observations of CO$_2$ mole fractions from a global observing network. The total carbon fluxes $F(x,y,t)$ for each region $r$ (defined by longitude $x$ and latitude $y$) and each time step ($t$) are represented by:

$$F(x,y,t) = \lambda_r \cdot F_{bio}(x,y,t) + \lambda_r \cdot F_{oce}(x,y,t) + F_{fossil}(x,y,t) + F_{fire}(x,y,t) \qquad (2)$$

The scaling vectors ($\lambda_r$) multiply $F_{bio}$ and $F_{oce}$, which are pre-calculated space-time patterns obtained from biosphere and ocean models (prior fluxes). Fossil fuel ($F_{fossil}$) and biomass burning ($F_{fire}$) emissions are not scaled/optimized. The monthly mean prior ocean fluxes in CTE2016 are from the ocean inversion by Jacobson et al. (2007). Earlier versions of CarbonTracker used prior biosphere and fire carbon fluxes from the CASA-GFED2 system (van der Werf et al., 2010). In CTE2016 this has

been replaced by the SiBCASA-GFED4 model (van der Velde et al., 2014). SiBCASA-GFED4 provides net carbon fluxes for the dominant vegetation type in each 1°x1° grid box globally for every 3 hours. Daily fire emissions are included in these biosphere model calculations based on satellite observed burned area (Giglio et al., 2013). The seasonal development of vegetation is scaled with the satellite observed greenness (Normalized Difference Vegetation Index (NDVI)) and absorption of radiation (fPAR). The fossil fuel emissions are from the EDGAR4.2 Database (2011), together with worldwide country-

and sector-specific time profiles derived by the Institute for Energy Economics and the Rational Use of Energy (IER) from the University of Stuttgart and constructed for the CARBONES project (http://www.carbones.eu/). The global total fossil fuel emissions are scaled with different regional annual trends for each continent to global totals as used in the global carbon budget (Le Quéré et al., 2016) of the Global Carbon Project (GCP).

These prior fluxes are transported with the TM5 transport model (Krol et al., 2005) on a global resolution of 3°x2° with zoom

region of 1°x1° over Europe and North America. TM5 uses meteorological driver data from the ERA-interim re-analysis (Dee et al., 2011) from the European Centre for Medium-Range Weather Forecasting (ECMWF). The convective entrainment and





detrainment fluxes are obtained directly from the ERA-interim data, whereas in earlier versions we used the Tiedtke convection scheme (Tiedtke, 1989). The resulting $CO_2$ mole fractions are compared to atmospheric $CO_2$ observations and their differences are minimized using the ensemble Kalman smoother, by adjusting the flux scaling vectors ($\lambda_r$) resulting in optimized posterior fluxes. The $CO_2$ observations are from the ObsPack product: GLOBALVIEWplus v2.1 (ObsPack, 2016). CTE2016 assimilates discrete (flask) samples as well as hourly values for well mixed conditions (afternoon hours for most locations, and nighttime hours for mountain locations).

The current setup of CarbonTracker Europe for $CO_2$ (CTE2016) has several differences compared to the current version of CarbonTracker at NOAA (CT2015). We document here the most important differences:

- CTE2016 uses CTDAS, CT2015 uses the implementation in TM5.

- CTE2016 uses two zoom regions in TM5 (over both North America and Europe), CT2015 uses a zoom over North America.

- CT2015 applies a larger a-priori flux uncertainty on land regions.

- CTE2016 uses the gridded state vector (Sect. 3.2), CT2015 uses the ecoregion state vector.

- CTE2016 and CT2015 use different prior fluxes for biosphere, ocean, fires and fossil fuels.

- CTE2016 and CT2015 use different subsets of $CO_2$ observations.

## 3.2 The gridded $CO_2$ state vector

Previous releases of CarbonTracker applied the same scaling factor for the biosphere fluxes ($\lambda_r$) to all grid boxes that share the same "ecoregion" type, which means they have a similar dominant land-cover type within a broader continental region (e.g. European Croplands). The land-cover types are defined by the Olson ecosystem classification (Olson et al., 2002), and the continental regions follow the TransCom definitions (Gurney et al., 2002). This approach implies that errors in the pre-calculated biospheric fluxes are fully correlated over the ecoregion, and adjustments needed to match atmospheric $CO_2$ mole fractions must be applied to all grid boxes of that ecoregion (proportional to the magnitude of the flux because of the linear scaling). Although this might be realistic within the context of the biosphere model that uses the same parameterizations for the same land-use types, this assumption can be questioned for actual carbon fluxes. Especially when ecosystems are geographically far apart (such as coniferous forests along the east and west coast of boreal North America), their responses to similar weather forcings might be quite different because of differences in e.g. age structure, or management regime.

A more realistic alternative is to assume no error correlations in the biosphere fluxes over space, an approach supported by independent research based on observations (Chevallier et al., 2010). However, since the density of the observing network does not allow each ecosystem in the world to be monitored and optimized independently, many other data assimilation systems assume that correlations between regions decay exponentially as a function of distance. This correlation length scale is chosen mostly based on practical considerations, and can vary from a few 100 km to more than 1000 km (e.g. Chevallier et al., 2010;





Rödenbeck et al., 2003; Basu et al., 2013). Effectively, this correlation strongly reduces the number of degrees of freedom in the covariance matrix ($\mathbf{P}^b$) of the scaling factors, balancing it with the number of observations. For instance, a gridded state vector for land fluxes at 5°x4° resolution has around 1000 land grid boxes, but only about 60 degrees of freedom when using a length scale of 1000 km (Peylin et al., 2013).

In CTDAS, we adopted this approach, enhanced with ecoregion information through the covariance, and implemented a gridded state vector for the Northern Hemisphere land regions on 1°x1° resolution. We still apply the region-based state vector to all ocean regions as well as the Southern Hemisphere ecoregions. To manage the degrees of freedom we use this approach only for the land TransCom regions of the Northern Hemisphere which are best constrained by observations, and we furthermore use variable length scales reflecting this observation network density. Moreover, in TransCom regions with a

gridded state vector we limit the correlations to exist only between grid boxes within the same Olson ecoregion, such that a priori errors in forest fluxes do not correlate with errors in crop fluxes even if they are dominant in neighboring grid boxes. The chosen prior standard deviation ($\sigma$) is 80% on land parameters, and 40% on ocean parameters, reflecting more prior confidence in the ocean fluxes than in terrestrial fluxes, because of the lower variability and larger homogeneity of the ocean fluxes. The maximum covariance is therefore 0.64 ($\sigma^2$) for land parameters. The structure of the new gridded state vector is summarized

in Table 1, showing a total number of 9835 scaling factors to be estimated each week, with close to 1100 degrees of freedom. An example of the covariance for a specific grid box in the European Conifer Forest region is given in Fig. 2.

    Within the new CTDAS system, the implementation of this new gridded state vector required the creation of: (1) a new global map that numbers each 1°x1° grid box according to its associated state vector element (N=1,...,9835), and (2) an a priori covariance matrix for this new state vector, (3) a new DaSystem class (see Sect. 2.3) that defined the state vector size for this

new configuration, and finally (4) a new StateVector class (GriddedStateVector) which inherited all methods from the base class StateVector (see Sect. 2.3), and in addition had modified methods to efficiently read the covariances and create ensemble members. This implementation is also flexible and can be used easily in other applications with different setups of the state vector (see Sect. 4).

### 3.3   CTE2016 results

We have started providing annual releases of the carbon flux estimates from CarbonTracker Europe since 2013. The current version is CTE2016 and includes carbon flux estimates for 2001-2015 (Sect. 3). CTE2016 uses the gridded state vector (Sect. 3.2). Other general details of the setup and e.g. prior fluxes are described in Sect. 3.1. Carbon fluxes are estimated for the period 2001-2015 and are shown annually for the global scale in Fig. 3. This figure shows the imposed fossil fuel and biomass burning emissions and the resulting ocean and land sinks. The natural $CO_2$ sinks show considerable interannual variability, mainly due

to climatic differences between the years. Since the land and ocean sinks are calculated from the emissions and the observed atmospheric $CO_2$ mole fractions, they reflect the interannual variability in the atmospheric growth rate. Figure 3 also shows the comparison of the total fluxes estimated by CTE2016 with the global atmospheric $CO_2$ growth rate as observed at background sites from the NOAA ESRL network (Dlugokencky and Tans, 2017). The growth rates are converted from ppm/yr to PgC/yr using 2.12 PgC per ppm following Prather et al. (2012). The total fluxes from CTE2016 match the observed atmospheric





growth rate well. The remaining differences reflect differences due to observation sites included in either the data assimilation or the calculation of the global growth rate.

The fossil fuel emissions increased from 6.8 PgC/yr in 2001 to 9.8 PgC/yr in 2015. The fossil fuel emissions in 2014 and 2015 are almost equal, but the 2015 atmospheric growth rate of 2.98±0.09 ppm/yr is much higher, compared to 1.99±0.09

5  ppm/yr in 2014. As shown in Fig. 3, CTE2016 assigns this anomaly to a smaller uptake by the biosphere, and in a lesser extent to a smaller ocean uptake. Biomass burning emissions have also slightly increased between 2014 and 2015.

Over the period 2001-2015, especially 2011 and 2014 stand out with high land uptake, and the carbon sinks in 2002, 2003 and 2005 were relatively low (Fig. 3). Figure 4 shows the annual development of the cumulative anomalies of the natural carbon fluxes (biosphere and ocean sinks, and the emissions from biomass burning). These anomalies are the difference to

the 2001-2015 mean. In 2011 and 2014, the sinks were relatively larger throughout the year. The year 2015 had higher than average uptake in summer, but this effect was cancelled by a reduced uptake in the remainder of the year.

Both natural sinks show an increasing trend over the period 2001-2015. The average ocean sink slightly increased from −1.9±0.8 PgC/yr in 2001-2003 to −2.5±0.1 PgC/yr in 2013-2015. The average land sink (including biomass burning emissions) increased from −1.8±1.0 to −2.3±0.8 PgC/yr over the same time periods. Global maps of the ocean and biosphere fluxes

(including biomass burning emissions) for the prior and posterior estimates averaged over the 2001-2015 period are shown in Fig 5. The average posterior biosphere sink (excluding biomass burning emissions) over 2001-2015 of −3.8 PgC/yr is larger compared to the prior estimate of −2.4 PgC/yr. The sink especially increases in the Northern Hemisphere. The average ocean sink of −2.3 PgC/yr is lower than the prior estimate of −2.7 PgC/yr, and also the trend in the ocean sink decreases from a prior estimate of −0.075 PgC/yr$^2$ to −0.044 PgC/yr$^2$.

Figure 6 shows the latitudinal distribution of the residuals of the simulated minus observed $CO_2$ mole fractions for all assimilated observations. With the exception of a few sites, the remaining biases are generally small and well below 1 ppm. The mean bias over all sites is 0.027±0.67 ppm, and the average of the absolute values of the biases is 0.31±0.59 ppm. There is a difference in the bias between the summer and winter, as the wintertime observations are generally better represented in CarbonTracker because of the lower variability in the $CO_2$ concentrations in winter. CTE2016 overestimates the $CO_2$ mole

fractions in the Northern Hemisphere summer and the average bias is 0.31±0.89 ppm. In the Northern Hemisphere winter this is -0.13±0.65 ppm.

Although CTE2016 optimizes fluxes on the global scale, carbon fluxes can also be estimated for smaller (eco)regions. Figure 7 shows the carbon sink of the European forest ecoregion over the period 2001-2015, together with the emissions from fossil fuels from the same region. Forest areas and human activities strongly overlap in Europe (on 1°x1° resolution). In most of the

30  years the forests take up carbon from the atmosphere and thereby partly compensate the emissions. There is some interannual variability and especially in years with droughts, like 2003 or 2010, the European forest carbon sink is reduced to zero. Other (eco)regions in Europe (specifically grasslands) are close to neutral, while croplands can add up to a small source in some years. Our forest carbon sink is in good agreement with e.g. Janssens et al. (2003), but not with the space-based estimate from Reuter et al. (2014), who find a larger sink in European forests.



### 3.4 Comparison of CTE2016 with previous releases

The first release of carbon flux estimates from CarbonTracker Europe (CTE) was in 2008 (CTE2008). Table 2 gives an overview of the different versions of CTE. Generally, the version IDs include the year in which the version is released and the simulation covers the years from 2001 until the year before the release date (e.g. CTE2008 covers 2001-2007, while CTE2013-OD is

5 an exception and covers 2001-2010). Simulations start in 2000, which is discarded and seen as a spin-up of the calculations. CTDAS (Sect. 2) was used for all versions from CTE2013-OD. Since 2014, CTE results have been included in the annual updates of the global carbon budget published by the Global Carbon Project (GCP) (CTE2014, CTE2015, CTE2016-FT in resp. Le Quéré et al., 2015b, a, 2016).

From version CTE2008 to version CTE2016, several changes have been implemented. Most of the prior fluxes have been

changed, except for the ocean prior fluxes, and the amount of observations and observational sites has increased. The most significant updates are 1) the implementation of the gridded state vector from version CTE2013 (Sect. 3.2), and 2) changes in the TM5 meteorology, including: a) changing from operational data from the European Centre for Medium-Range Weather Forecasting (ECMWF) to using era-interim re-analysis driver data (Dee et al., 2011), and b) the use of the convective entrainment and detrainment fluxes directly from ECMWF from version CTE2014, instead of using the previous Tiedtke convection

scheme.

The differences in the estimated natural carbon fluxes (ocean and biosphere including biomass burning emissions) between the different versions are shown in the left panel of Fig. 8 for selected regions for their overlapping period 2001-2007. The posterior uncertainty in CarbonTracker can be estimated by different approaches. The right panel includes the fluxes for a single region (Northern land) together with two options for the uncertainty estimate. The first option shows the internal error

based on the weekly posterior covariance matrix. A new prior covariance is included for each new week in the inversion, not taking into account information on the uncertainty (reduction) in the previous weeks. This results in a unrealistically large error estimate due to the absence of temporal correlation of the covariance. The advantage is that fluxes from different regions remain uncoupled in new weeks. Alternatively, the uncertainty of an inversion can be estimated by the range between estimates from several different realizations (e.g. Peylin et al., 2013). The second option in Fig 8 shows range between the seven versions

of CarbonTracker Europe. This is our preferred option and is also used in Peters et al. (2010) and van der Laan-Luijkx et al. (2015). The resulting carbon fluxes from these versions show differences based on choices made in their setups. In the most recent version CTE2016 we have updated the fossil fuel emissions over the total period 2000-2015 to match the total global emissions used in GCP (Sect. 3.1). These higher emissions lead to larger carbon sinks, especially in the Northern Hemisphere. Following from the uncertainty estimate taken as the range of the different versions, we can state that the change between

CTE2008–CTE2013 to CTE2014–CTE2016-FT has a significant effect on the resulting carbon flux estimates, which is a result of the used convective fluxes. The distribution of the sinks over the hemispheres shifted from North to Tropics and from the land to the oceans. With the updated convection, the land sink is especially decreased in the Northern Hemisphere, and the ocean sink is slightly increased in both the Northern and Southern Hemisphere.





## 4 Overview of applications using CTDAS

Besides global $CO_2$ fluxes as presented in Sect. 3, the CTDAS framework has also been used in several applications with focus on different regions or different greenhouse gases and related tracers. We developed a dedicated version of CTDAS focusing on the Amazon carbon balance: CT-SAM (van der Laan-Luijkx et al., 2015). With CT-SAM we found that the response

of the Amazon carbon balance to the 2010 drought was twofold: the biospheric uptake decreased and the emissions from biomass burning doubled. The total reduction of carbon uptake was 0.24–0.50 PgC/yr and turned the balance from carbon sink to source. We also developed a multi-tracer version of CTDAS including both $CO_2$ and $\delta^{13}CO_2$ (van der Velde, 2015; van der Velde et al., 2017). Using these combined signals together allowed optimization of both carbon fluxes and the isotope discrimination parameters. The results showed that isotope discrimination was decreased during severe droughts leading to an

increase in intrinsic water use efficiency of up to 25%.

CTDAS was also used to develop $CO_2$ data assimilations systems with a specific focus on Asia and China in particular. This region is highly relevant in the carbon cycle due to the large $CO_2$ emissions from fossil fuel combustion. Zhang et al. (2014a, b) showed that Chinese terrestrial ecosystems took up 0.33 PgC/yr on average during 2001–2010, thereby compensating approximately 20% of the total $CO_2$ emissions from fossil fuel combustion from China. For Asia in total, this effect is even larger:

during 2006–2010 the Asian terrestrial land $CO_2$ sink was $-1.56$ PgC/yr, which is about 37% of the Asian fossil fuel emissions ($+4.15$ PgC/yr). Jiang et al. (2016) suggest that the Chinese terrestrial $CO_2$ uptake is increasing over the past decades. This is also confirmed by Thompson et al. (2016), a study based on seven atmospheric inversions including CTE2014, which shows that the annual $CO_2$ sink in East Asia increased between 1996–2001 and 2008–2012 by 0.56 (0.30–0.81) PgC/yr, accounting for 35% of the increase in the global land biosphere sink.

CarbonTracker Europe results have been included in several studies focusing on different aspects of the carbon cycle. CTE2014 has e.g. been included in a study of the 2012 drought in the US (Wolf et al., 2016), where a warm spring increased biospheric carbon uptake, compensating for the reduction in carbon uptake in the following summer drought. In this analysis, it was also shown that the use of CTE2014 with the new gridded state vector and the 3 hourly resolution of the prior biosphere fluxes was better suited to detect anomalies in the timing of the start of the growing season, compared to CT2013B (NOAA).

Babenhauserheide et al. (2015) evaluate the differences between two data assimilation approaches for $CO_2$: the ensemble Kalman smoother approach of CTDAS and the TM5-4DVar method. Several aspects of the data assimilation are addressed including the choices made in the window length for CarbonTracker and sensitivity to observational coverage. The carbon flux estimates from both optimization methods show increasing agreement with observational density. The CarbonTracker approach was shown to result in a higher bias between the simulated and observed mole fractions in remote regions (e.g. South

Pole), given its 5 week assimilation window. On the other hand, the 4DVar method with its longer window is more susceptible to changes in observational coverage and has larger covariances between regions. Increasing CarbonTracker's window length to improve the bias at remote sites could also result in projection of fluxes in regions with limited observational coverage, specifically the tropics as demonstrated in van der Laan-Luijkx et al. (2015).



CTDAS is used at the Finnish Meteorological Institute (FMI) for the development of CarbonTracker Europe Methane (CTE-$CH_4$) and is used to perform global methane inversions (Tsuruta et al., 2015, 2016). Both anthropogenic and biosphere emissions of $CH_4$ are simultaneously constrained by global atmospheric $CH_4$ mole fraction observations. The mean global total emissions during 2000-2012 were estimated to be 516±51 Tg $CH_4$ per year of which about 60% are of anthropogenic origin

and 30% are biospheric. Emissions in the 2007–2012 period were on average 18 Tg $CH_4$ per year larger compared to the 2001–2006 period.

CTDAS has also been used for the optimization of transport properties of the underlying TM5 model using observations of $SF_6$ (van der Veen, 2013). Previous studies demonstrated that many models, including TM5, poorly simulate the $SF_6$ gradients between the Northern Hemisphere (NH) and Southern Hemisphere (SH), which is mainly controlled by transport across the

intertropical convergence zone (ITCZ). After lifting by the strong convective motions near the Tropics, $SF_6$–rich air from the NH can make its way into the SH through lateral outflow. Many models underestimate the efficiency of this process, as it is often not resolved numerically on the grid scales used for global modeling. As a result, the interhemispheric exchange time of these models is too slow, and gradients in $SF_6$ between the NH and SH are overestimated. Inversions with $SF_6$ improved the north-south transport of TM5 by accelerating its horizontal sub-grid scale transport in the convection scheme. The results were

used as an intermediate solution for the setup of TM5 in CTE2013 (indicated as newslopes in Table 2) before switching from the old Tiedtke convection scheme to using the convective fluxes directly from ECMWF.

All CTDAS applications mentioned above used TM5 as the observation operator and were applied to the global scale. Other applications on regional scales are currently being developed using different transport models. CTDAS-Lagrange (developed at University of Groningen) combines CTDAS with a high-resolution Lagrangian transport model, the Stochastic Time-Inverted

Lagrangian Transport model driven by the Weather Forecast and Research meteorological fields (WRF-STILT) (He et al., 2017). This system assimilates atmospheric observations of both $CO_2$ and COS to constrain gross primary production and ecosystem respiration for North America. Footprints for each $CO_2$ and COS observation are precalculated, making this a computationally more efficient method than using an Eulerian model. Resulting $CO_2$ flux estimates for North America in 2010 are comparable to estimates from CTE2014 and higher than those from CT2013B (NOAA) (He et al., 2017). A second regional

application focuses on Switzerland, and is developed at ETH Zürich. CTDAS is combined with the new tracer transport module of the regional numerical weather prediction model COSMO, and is used to estimate carbon fluxes in Switzerland, making use of $CO_2$ observations from four new measurement sites around Switzerland (Liu et al., 2017; Oney et al., 2015). The resulting $CO_2$ flux estimates match well with the bottom-up estimates.

## 5  Conclusions and outlook

We demonstrated the use of our new data assimilation framework: the CarbonTracker Data Assimilation Shell (CTDAS). This framework allows flexible setup of different components of the data assimilation system and can be used in a wide range of applications. We have shown the most recent developments for the CarbonTracker Europe $CO_2$ system: CTE2016, especially the implementation of the gridded state vector. We have shown results from CTE2016 on the global scale. Resulting flux



estimates and $CO_2$ mole fractions are available from http://www.carbontracker.eu. We will provide annual updates and in the near future these will also be made available through the ICOS Carbon Portal (http://www.icos-cp.eu).

Upcoming developments for CTDAS include e.g. the expansion with more options for regional and urban applications with the use of different transport models as observations operator. We are also evaluating the implementation of the new version of
TM5: TM5-mp (massive parallel). TM5-mp can be run parallel over grid cells instead of tracers and thereby offers the possibility to efficiently simulate the transport on global $1°x1°$ resolution. Other developments in TM5 include the implementation of online meteorology. We will furthermore focus on new options for optimization methods and covariance structure. We are studying methods to account for temporal correlation in the state covariance matrix. Also we will study the effects of using shorter data assimilation windows (e.g. Kang et al., 2012; Liu et al., 2012) on our resulting fluxes. Finally, we will also focus
on the European carbon balance by specifically re-evaluating the fluxes from croplands Combe (2016).

## 6   Code and data availability

The CTDAS code (current revision r1479) is included as supplement and is open access under GNU General Public License version 3. The actual CTDAS code is continuously updated and under version control (SVN) on a local server at Wageningen University & Research. Access can be granted after contacting the main developers. The documentation of the code (user
manual) prepared with SPHINX (see Sect. 2.5) is available at: http://www.carbontracker.eu/ctdas. The input data used for CTDAS depends per application, and can be made available upon request.

*Author contributions.*  I.T.v.d.L.-L. and W.P. prepared the manuscript with contributions from all co-authors. W.P. developed CTDAS, with contributions from the other authors. CTE2008, CTE2013-OD, CTE2013, CTE2014, CTE2015 and CTE2016-FT, CTE2016 simulations were performed by I.T.v.d.L.-L. and W.P..

*Acknowledgements.*  The authors thank the contributing laboratories for providing the atmospheric $CO_2$ observations from a global network of measurement sites through ObsPack products GLOBALVIEWplus version 1.0, 2.0 and 2.1, NRT v3.0, and previous prototypes. We acknowledge the NOAA CarbonTracker team, specifically Andy Jacobson, for the fruitful collaboration. CTE2016 simulations (and previous releases) have been performed using a grant for computing time (SH-312-14) from the Netherlands Organization for Scientific Research (NWO). I.T.v.d.L.-L. was funded by OCW/NWO for ICOS-NL, and is currently funded by a NWO Veni grant (016.Veni.171.095). W.P. is
supported by an ERC consolidator grant (649087). Part of the results included were supported by the GEOCARBON project (EU FP7/2007-2013, grant agreement 283080).



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



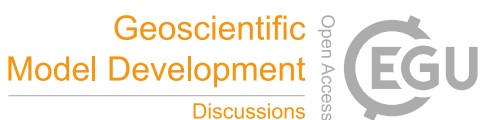

**Table 1.** Gridded state vector setup per TransCom land region and for global ocean regions, including details on the covariance, length scale, number of parameters and degrees of freedom (d.o.f.).

| TransCom region | State vector | Covariance | Length Scale | Parameters | d.o.f. |
|---|---|---|---|---|---|
| North America Boreal | gridded | within ecoregions | 300 km | 1865 | 184 |
| North America Temperate | gridded | within ecoregions | 300 km | 1213 | 242 |
| South America Tropical | ecoregion | across ecoregions | - | 19 | 3.2 |
| South America Temperate | ecoregion | across ecoregions | - | 19 | 2.9 |
| Northern Africa | ecoregion | across ecoregions | - | 19 | 3.2 |
| Southern Africa | ecoregion | across ecoregions | - | 19 | 2.5 |
| Eurasia Boreal | gridded | within ecoregions | 1000 km | 2396 | 63 |
| Eurasia Temperate | gridded | within ecoregions | 1000 km | 2631 | 129 |
| Tropical Asia | ecoregion | across ecoregions | - | 19 | 2.5 |
| Australia | ecoregion | across ecoregions | - | 19 | 3.4 |
| Europe | gridded | within ecoregions | 200 km | 1585 | 435 |
| Oceans | ocean regions | across ocean regions | - | 30 | 7 |
| Ice (not optimized) | - | - | - | 1 | - |





**Table 2.** Version information for CarbonTracker Europe simulations, with details on used setup including prior fluxes, observations, meteorological data and TM5 setup.

| Version ID | State vector | Biosphere/Fire[a] | Fossil fuel | Observations | TM5/Meteo |
|---|---|---|---|---|---|
| CTE2016 | Gridded | SiBCASA-GFED4 3 hourly | Carbones + GCP[b] | ObsPack GVplus2.1 | EI-convec |
| CTE2016-FT[c] | Gridded | SiBCASA-GFED4 3 hourly | Carbones + GCP[d] | ObsPack GVplus1.0 + NRT | EI-convec |
| CTE2015 | Gridded | SiBCASA-GFED4 3 hourly | Carbones + GCP[e] | ObsPack GVplus1.0 | EI-convec |
| CTE2014 | Gridded | SiBCASA-GFED4 monthly | Carbones | ObsPack Prototype 1.0.4b | EI-convec |
| CTE2013 | Gridded | SiBCASA-GFED4 monthly | Carbones | ObsPack Prototype 1.0.3 | EI-newslopes[f] |
| CTE2013-OD[g] | Ecoregion | SiBCASA-GFED4 monthly | Miller + IER | ObsPack Prototype 1.0.3 | OD |
| CTE2008 | Ecoregion | CASA-GFED2 monthly | Miller | Pre-ObsPack & CarboEurope | OD (glb6°x4°)[h] |

[a]Time resolution for the biosphere fluxes is either 3 hourly or monthly, while fire emissions are daily.

[b]Global total fossil fuel emissions are scaled to the values included in the global carbon budget (Le Quéré et al., 2016) of the Global Carbon Project (GCP) for 2000-2015.

[c]FT stands for Fast-Track, since inclusion in Le Quéré et al. (2016) required completion of the analysis before all observations became available.

[d]Same as a, but using values from Le Quéré et al. (2015a) for 2010-2014, and Le Quéré et al. (2016) for 2015.

[e]Same as c, for 2010-2014.

[f]Newslopes refers to the updated slopes scheme in TM5, based on simulations with $SF_6$.

[g]Irregular version ID: covers 2001-2010.

[h]The standard setup for TM5 is with a global spatial resolution of 3°x2° and 2 zoom regions over Europe and North America of 1°x1°. Only for CTE2008 we used a global resolution of 6°x4° with a two-way nested zoom over Europe of 3°x2° and 1°x1°.





**CTDAS classes**

Control classes:
1. CycleControl*
2. DaSystem*
3. Platform

Complementary classes:
4. StateVector
5. Observations*
6. ObservationOperator*
7. Optimizer

**Figure 1.** Overview of the 7 Python classes that comprise the CTDAS code base. Asterisks indicate passing information to the code through external run-control files.





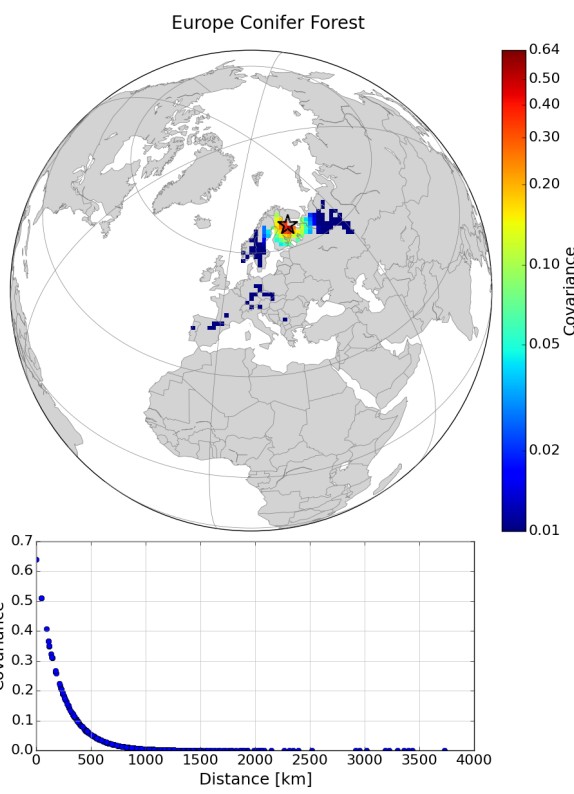

**Figure 2.** Error correlation in the gridded state vector setup for a specific grid box (indicated by the black star) in the European Conifer Forest region (with length scale 200 km) with the other grid boxes in that region (top panel) and versus distance (bottom panel).





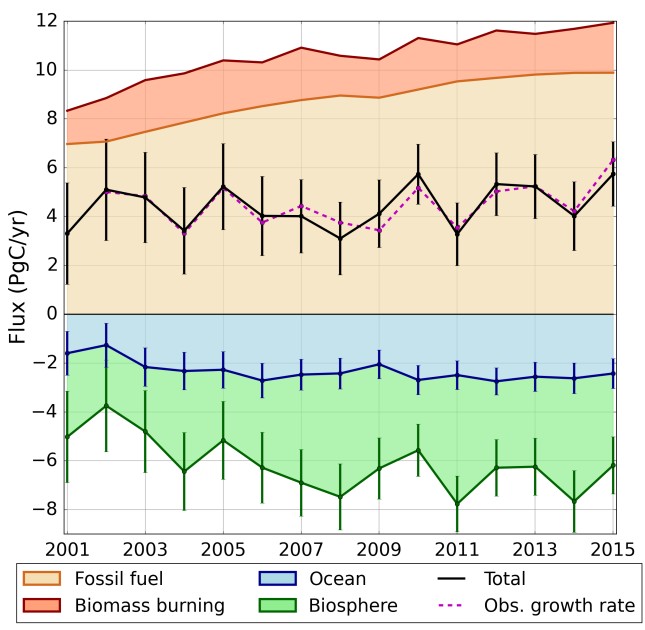

**Figure 3.** Global annual carbon balance estimated with CTE2016 for the period 2001-2015. Global ocean (blue) and biosphere (green) sinks are indicated as negative values and represent uptake from the atmosphere. The error bars represent the annual 1 $\sigma$ uncertainty, based on the average weekly covariances (more information on the error estimates in CarbonTracker in given in Sect. 3.4). Fossil fuel (orange) and biomass burning (red) emissions are not optimized. The total flux (black line) is the sum of the four components. The observed global annual atmospheric $CO_2$ growth rate from the NOAA network (dashed magenta line) was converted from ppm/year using a conversion factor of 2.12 PgC per ppm (Prather et al., 2012).





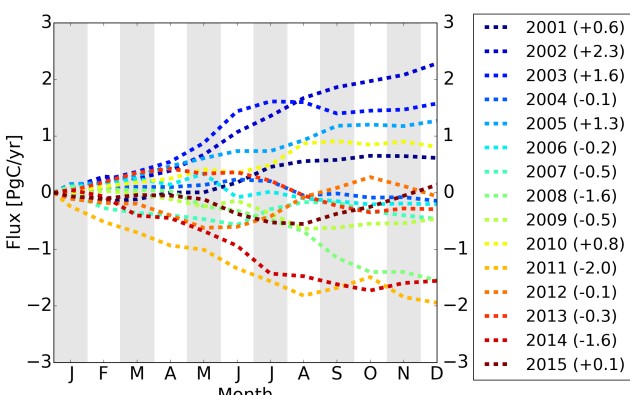

**Figure 4.** Monthly development of the cumulative annual anomalies in the global natural carbon fluxes (biosphere and ocean sink and biomass burning emissions). Anomalies are calculated from the mean over 2001-2015 for each year, thereby removing the average seasonal cycle. Negative numbers indicate years with larger than average uptake and positive numbers represent years with smaller than average uptake.





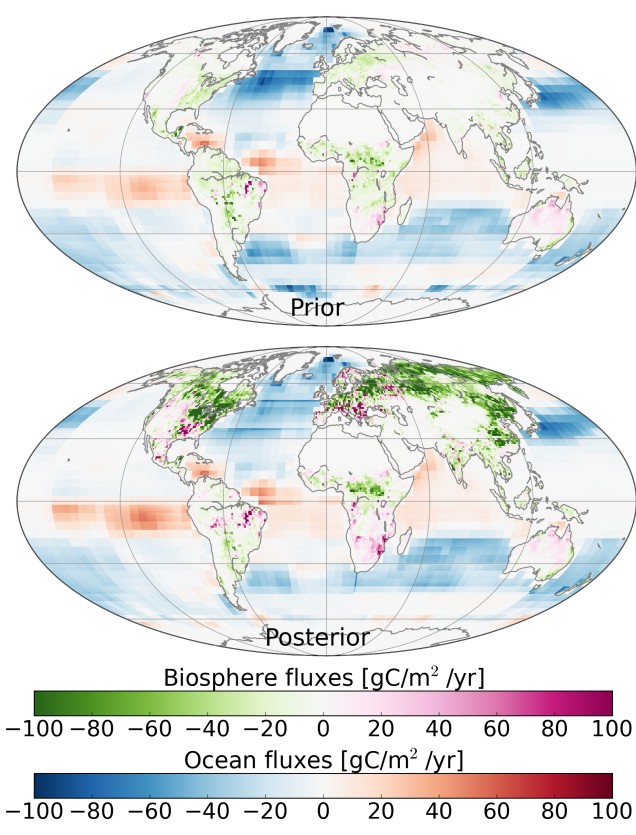

**Figure 5.** Global fluxes averaged over 2001-2015 for the prior estimate (top) and posterior/optimized estimates (bottom). Ocean and biosphere fluxes are shown on different color scales in gC/m$^2$/yr. Biosphere fluxes include imposed biomass burning emissions.





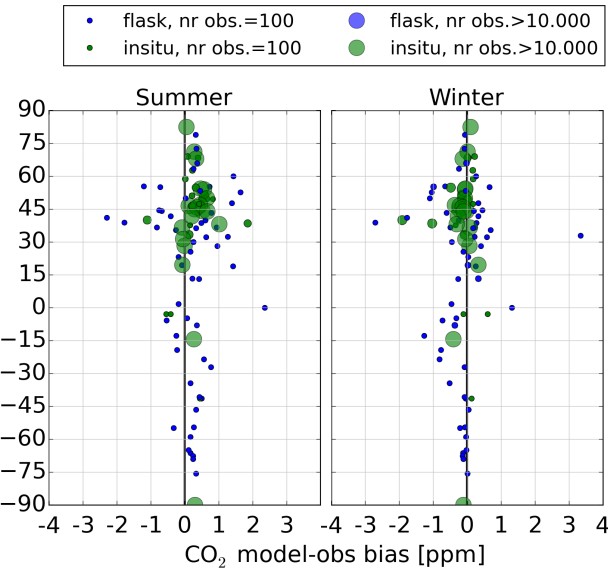

**Figure 6.** Latitudinal distribution of the average posterior residuals per site over the period 2001-2015, for Northern Hemisphere summer (left) and winter (right). The residuals are calculated as the difference of the simulated minus observed $CO_2$ mole fractions for all assimilated observations. First guess estimates of modeled $CO_2$ mole fractions which are more than three times the model-data mismatch are excluded.





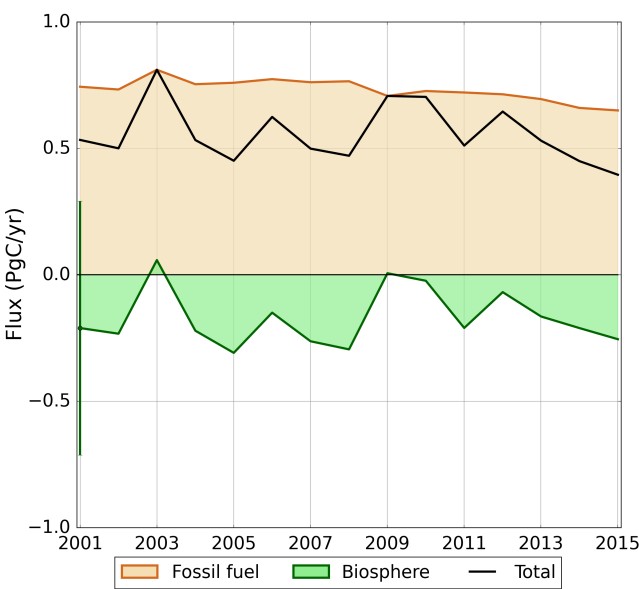

**Figure 7.** Annual carbon balance for European forests estimated with CTE2016 for the period 2001-2015. The biosphere (green) sink is shown together with the fossil fuel (orange) emission from the same region. The error bar represents the annual 1 $\sigma$ uncertainty, based on the average weekly covariances, and is shown only for 2001 for clarity (more information on the error estimates in CarbonTracker in given in Sect. 3.4). The total flux (black line) is the sum of the components.





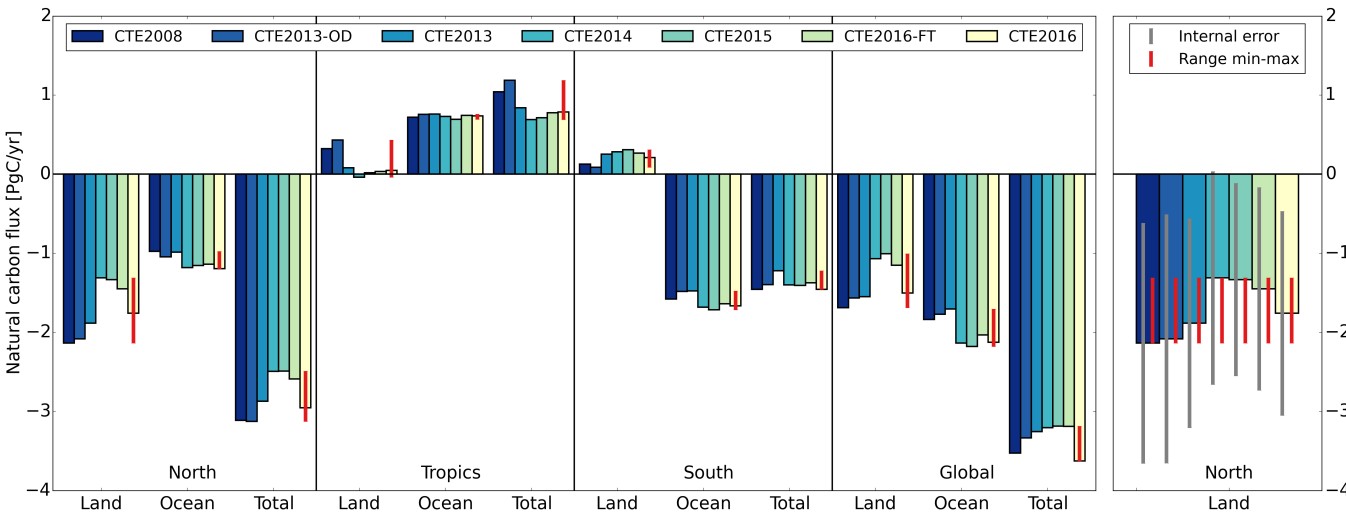

**Figure 8.** Average natural carbon flux estimates for different CTE versions for selected regions for the period 2001-2007 (left panel). The fluxes are the sum of the biosphere and ocean fluxes and biomass burning emissions. Two alternative uncertainty estimates are given for a selected region (right panel). The first is the internal error based on the average weekly posterior covariances (n=418), while the second is representing the range between the different realizations of the inversion (n=7). The second option is applied as the posterior uncertainty estimate per region in the left panel.