# Peer review of "The CarbonTracker Data Assimilation Shell (CTDAS) v1.0: implementation and global carbon balance 2001–2015"

_Geoscientific Model Development, 2017_

## Referee Comment (RC1) · Anonymous Referee #1 · 9 Apr 2017

**1   General comments**

This paper presents the new developments of CarbonTracker, a well known data assimilation system used to estimate carbon fluxes. These developments are possible thanks to the implementation of a new shell based on python. The presentation of this new shell is clear and well structured. The results are also well presented, comparing the system with other versions and other systems and highlighting the main achievements and challenges, as well as future plans for development. The technical aspects of the shell and the strategy to allow a flexible use of different components (e.g. observation operator, data assimilation methodology) are very interesting and relevant in

this fast evolving field of carbon cycle data assimilation. I recommend this paper to be accepted with minor corrections (see specific comments below).

**2  Specific comments**

- Page 3, lines 4-6: Include the sections relevant to each of the aspects addressed in the paper.

- Page 11: The posterior fluxes are net fluxes, therefore any change in the net sink can only be interpreted as a change in the net uptake. I would advice to replace 'uptake' by 'net uptake' and 'sink' by 'net sink'.

- Page 11, line 22: It would be interesting to show the standard deviation of the bias, as it reflects the capability of the posterior fluxes to represent the spatial patterns in the fluxes, i.e. the inter-station bias.

- Page 11, line 24: The winter transport is also easier to simulate (with large-scale planetary waves) than the smaller-scale convective transport during summer.

- Page 11, line 33: remove 'e.g.' before Janssens.

- Page 14, line 5: replace 'biospheric' with 'biogenic'.

- Page 22, Table 2: provide a reference for all the prior fluxes.

- Page 28, Figure 6: The last line is not clear.

---

## Referee Comment (RC2) · Anonymous Referee #2 · 21 Apr 2017

**General comments**

The authors present a new design of the data assimilation code named "Carbon-Tracker". The paper is interesting at first glance and well written, but the scientific content is rather shallow. The editor has a much better view than me about what can be published in GMD, but as a reader I feel rather frustrated. The core of the paper is structured into four parts. The first one is like "CarbonTracker meets Python and adopts classes and modularity". This change certainly represented a large amount of thinking and work, but the use of Python which is described is quite basic and common. My codes are mostly in Python and are structured the same way, even for the documentation, and my colleagues do roughly the same. Again, this is not a judgement about the technical value of the work, or about the involvement of the developers, but rather a judgement about its meaning for external readers. The second part is about recent results obtained for $CO_2$ and describes recent updates of the configuration. There are interesting parts (in particular the comparison between the successive product releases) but that may not go far enough. I could not find any information about the way the error statistics are cycled from one window to the next, or about the way temporal correlations are handled within an assimilation cycle (actually the last lines of the paper suggest that they are not handled at all, which is surprising), or about the global prior error budget, or about the ensemble size in the gridded state vector configuration (given the curse of dimensionality), etc. Some of the results also seem to have already appeared in Le Quéré et al. (2016). Posterior errors are shown but are immediately discredited (p. 12, l. 21-22), which suggests a major gap in the new shell. The third part is an overview of applications: it shows that the authors have nicely structured a community, but is there anything scientific that the reader should take from it? The fourth part is a short list of planned developments. I would recommend that the authors put more scientific material in their paper before it is published in GMD.

**Detailed comments**

- p. 1, l. 11,"We show…": what is the difference with the CTE material and associated conclusions displayed in Le Quéré et al. (2016)? How robust is this result (can the atmospheric data properly separate between land and ocean fluxes?)?

- p. 1, l. 12-13: is this really news (that forests are the dominant sink in Europe and that drought reduces it)?

- p. 1, l. 13: do the authors suggest that the historical version was not versatile and could not allow such applications? I know several large Fortran codes that

still had many applications despite horrible coding.

- p. 2, l. 11: computational time has nothing to do with the number of code lines.

- p. 3, l. 15 and 17: "error" is missing before "covariance" at both places.

- p. 3, l. 19: if the system is robust for Europe and Northern America only, why are results for other parts of the globe shown (e.g. Fig. 8)?

- p. 4, l. 3: from a quick check of the ECMWF web site, the situation of OOPS at ECMWF seems to be less advanced than what is suggested here (http://www.ecmwf.int/sites/default/files/elibrary/2017/17179-strategy-data-assimilation.pdf).

- p. 10, l. 10: "Olson ecoregion" is not a standard expression.

- p. 10, l. 34: I could not find the 2.12 factor in Prather et al. (2012). The reference may be wrong.

- p. 11, l. 1: "well" should be quantified. What is the scientific meaning of the error bars in the figure if they are not properly computed (also for Fig. 7)?

- p. 11, l. 1-2: the authors forget the role of transport model errors and the assumptions behind the NOAA estimates.

- p. 12, l.22-23: how can this feature be an advantage? I would think otherwise.

- p. 12, l. 23: comparing a range with a standard deviation is not trivial. How is this done? Is the range assumed to represent 4 sigmas, 6 sigmas, . . .?

- p. 12, l. 24: this statement is valid only under the requirement that the realizations are of the same quality level.

- p. 13, l. 29: this result does not seem consistent with the plan to further shorten the assimilation window (p. 15, l.9).

- p. 13, l. 31: the statement is intriguing and I could not find its origin in the Babenhauserheide et al paper. In its section 5.1.1, the latter paper discusses rejection and error assignment issues rather than optimization methods per se.

- p. 13, l. 33: the statement seems to be too trivial for a "demonstration". "Illustrate" would be better, or am I missing something?

- p. 14, l. 30: why did the use of the new Python shell need to be demonstrated in the first place?

---

## Author Comment (AC1) · 12 Jun 2017

**Reply to reviewers on "The CarbonTracker Data Assimilation Shell (CTDAS) v1.0: implementation and global carbon balance 2001–2015"**

The authors would like to thank both reviewers for their efforts to review this manuscript. The suggestions are appreciated and incorporated in the revised version of the manuscript. Below we address each point raised by the reviewers separately.
* * *
**Referee #1**

**1 General comments**
*This paper presents the new developments of CarbonTracker, a well known data assimilation system used to estimate carbon fluxes. These developments are possible thanks to the implementation of a new shell based on python. The presentation of this new shell is clear and well structured. The results are also well presented, comparing the system with other versions and other systems and highlighting the main achievements and challenges, as well as future plans for development. The technical aspects of the shell and the strategy to allow a flexible use of different components (e.g. observation operator, data assimilation methodology) are very interesting and relevant in this fast evolving field of carbon cycle data assimilation. I recommend this paper to be accepted with minor corrections (see specific comments below).*
**Authors**: Many thanks for this positive assessment.

**2 Specific comments**
• *Page 3, lines 4-6: Include the sections relevant to each of the aspects addressed in the paper.*
  **Authors**: Section numbers have been included.

• *Page 11: The posterior fluxes are net fluxes, therefore any change in the net sink can only be interpreted as a change in the net uptake. I would advice to replace 'uptake' by 'net uptake' and 'sink' by 'net sink'.*
  **Authors**: We agree this is more clear and have added 'net' at several places throughout the manuscript.

• *Page 11, line 22: It would be interesting to show the standard deviation of the bias, as it reflects the capability of the posterior fluxes to represent the spatial patterns in the fluxes, i.e. the inter-station bias.*
  **Authors**: We have added 2 panels to figure 6 to include the standard deviation of the biases.

• *Page 11, line 24: The winter transport is also easier to simulate (with large-scale planetary waves) than the smaller-scale convective transport during summer.*
  **Authors**: Added this additional explanation.

• *Page 11, line 33: remove 'e.g.' before Janssens.*
  **Authors**: Done.

- *Page 14, line 5: replace 'biospheric' with 'biogenic'.*
  **Authors**: Done.

- *Page 22, Table 2: provide a reference for all the prior fluxes.*
  **Authors**: Done.

- *Page 28, Figure 6: The last line is not clear.*
  **Authors**: The sentence has been rewritten.
* * *
**Referee #2**

**General comments**

*The authors present a new design of the data assimilation code named "Carbon-Tracker". The paper is interesting at first glance and well written, but the scientific content is rather shallow. The editor has a much better view than me about what can be published in GMD, but as a reader I feel rather frustrated. The core of the paper is structured into four parts. The first one is like "CarbonTracker meets Python and adopts classes and modularity". This change certainly represented a large amount of thinking and work, but the use of Python which is described is quite basic and common. My codes are mostly in Python and are structured the same way, even for the documentation, and my colleagues do roughly the same. Again, this is not a judgement about the technical value of the work, or about the involvement of the developers, but rather a judgement about its meaning for external readers. The second part is about recent results obtained for $CO_2$ and describes recent updates of the configuration. There are interesting parts (in particular the comparison between the successive product releases) but that may not go far enough. I could not find any information about the way the error statistics are cycled from one window to the next, or about the way temporal correlations are handled within an assimilation cycle (actually the last lines of the paper suggest that they are not handled at all, which is surprising), or about the global prior error budget, or about the ensemble size in the gridded state vector configuration (given the curse of dimensionality), etc. Some of the results also seem to have already appeared in Le Quéré et al. (2016). Posterior errors are shown but are immediately discredited (p. 12, l. 21-22), which suggests a major gap in the new shell. The third part is an overview of applications: it shows that the authors have nicely structured a community, but is there anything scientific that the reader should take from it? The fourth part is a short list of planned developments. I would recommend that the authors put more scientific material in their paper before it is published in GMD.*

**Authors**: The reviewer raises several main points here, which unfortunately seem to be based on wrong expectations of a GMD 'model description paper' manuscript type. With this manuscript we exactly had the goal to describe our modeling framework. In comparison to the previous version of CarbonTracker (which was integrated in the TM5 transport model's code), there have been several substantial changes which especially allow CTDAS to be applied more easily to a much wider range of applications than the original version. We feel it

is important to document these updates, together with the current version of the code, so that it is openly available to anyone interested, and can serve as a reference in publications that have their focus on the scientific results using CTDAS.

GMD is especially appropriate for this purpose, specifically in the form of a 'model description paper' manuscript type. The instructions on the GMD webpage include e.g. that GMD has a wide definition of the term model and can range from 'comprehensive descriptions of numerical models' to e.g. 'spreadsheet-based models and box models' and includes also e.g. 'coupling frameworks and software toolboxes with a geoscientific application'. This definition covers CTDAS very well in our opinion.

The instructions on the GMD website furthermore specifically ask for a contextualization of the model description, in the form of e.g. a 'scope of applicability'. We have included all current applications in Section 4 to demonstrate the applications of CTDAS. GMD also asks for 'examples of model output'. We have included Section 3 as the current main application of CTDAS which is CarbonTracker Europe, and serves as a reference to new developments in our general $CO_2$ application, which is widely used in the carbon cycle community (e.g. Le Quéré et al., 2016). The new developments since 2010 (e.g. the use of the gridded state vector) had not yet been published integrally and this manuscript is now an up-to-date documentation of the current setup.

We understand that this reviewer would have liked to see 'more scientific material'. We have found that combining the description and documentation of CTDAS with scientific results (e.g. van der Laan-Luijkx et al., 2015) does not fit in a single paper, and more science will definitely follow in additional manuscripts of which some are already hinted to in Section 4 as publications in preparation.

**Replies to the specific questions in the above 'general comments':**
*I could not find any information about the way the error statistics are cycled from one window to the next, or about the way temporal correlations are handled within an assimilation cycle (actually the last lines of the paper suggest that they are not handled at all, which is surprising),*
**Authors**: We have chosen to focus this manuscript on the changes compared to the previous version of CarbonTracker as integrated in the TM5 transport model's code (which has been documented extensively in Peters et al., 2005, 2007 and 2010). The implementation of the data assimilation technique in the form of the Ensemble Kalman smoother has not changed in principle, it has just been translated to the Python version of the new code. The propagation of the errors and temporal correlations have not changed in this new version. The prior scaling factors are the average of the prior scaling factors (1.0) and the optimized scaling factors from the previous two time steps: $\lambda_t^b = (\lambda_{t-2}^a + \lambda_{t-1}^a + \lambda^p)/3.0$, as shown in Peters et al. (2007). This information has been added to Section 3.1.

*or about the global prior error budget,*
**Authors**: Section 3.2 includes that the standard prior standard deviation is 80% for land parameters and 40% for ocean parameters. The prior carbon budget is shown below for reference in comparison to the optimized budget as in Figure 3 in Figure R1 below.

[Figure]

**Figure R1**. Global annual carbon balance estimated with CTE2016 for the period 2001-2015. Prior fluxes are shown on the left panel and optimized fluxes are shown on the right panel. The error bars represent the annual 1 σ uncertainty, based on the average weekly covariances.

*or about the ensemble size in the gridded state vector configuration (given the curse of dimensionality), etc.*
**Authors**: The number of ensemble members is still 150. It was included in Section 2.1 and this number is now repeated in Section 3.1.

*Some of the results also seem to have already appeared in Le Quéré et al. (2016).*
**Authors**: CTE is one of many contributions to the large community effort published by Le Quéré et al. (2016). Le Quéré et al. (2016) includes the results from CTE2016-FT for the total land sink (not split out for net biosphere exchange and biomass burning), and for the distribution of the total fluxes over the hemispheres. These aspects are not repeated in our manuscript, and we provide a more detailed overview of the results from the more recent version CTE2016.

*Posterior errors are shown but are immediately discredited (p. 12, l. 21-22), which suggests a major gap in the new shell.*
**Authors**: Posterior errors have been 'discredited' since the first version of CarbonTracker, and this is not related to the introduction of the new CTDAS shell in this paper. The first publication on CarbonTracker (Peters et al., 2005) already states that the across-model spread or external uncertainty has more meaning than the formal posterior uncertainty for a single inversion, as repeated also by later publications (e.g. Peylin et al., 2013). This is because meaningful propagation of covariations through time and space requires a dynamical model for the state vector, in addition to a large observation network to constrain the covariances. Both are missing in virtually every atmospheric inverse modeling framework currently in use (the exception being the pseudo-data applications of Kang et al. (2012)). CTDAS is additionally challenged by its short temporal window, which precludes the possibility to derive annual mean uncertainties from its covariance matrix, which is possible for some other techniques (e.g. Chatterjee and Michalak (2013) and Chevallier, et al. (2010)).

**Detailed comments**

- *p. 1, l. 11,"We show...": what is the difference with the CTE material and associated conclusions displayed in Le Quéré et al. (2016)? How robust is this result (can the atmospheric data properly separate between land and ocean fluxes?)?*
  **Authors**: Le Quéré et al. (2016) do not include conclusions based on CTE alone. Its main carbon budget results are not based on the inversions. The inversion results are included especially to derive year-to-year variability in the total land fluxes and for the spatial breakdown of the total land and ocean fluxes. The conclusions in our manuscript are based on the CTE results alone. The robustness of the CTE estimates is discussed in Section 3.4 and in Figure 8 of the manuscript.

- *p. 1, l. 12-13: is this really news (that forests are the dominant sink in Europe and that drought reduces it)?*
  **Authors**: We agree this is a very general statement and have added more detailed information on the CTE2016 estimate of the European forest carbon sink.

- *p. 1, l. 13: do the authors suggest that the historical version was not versatile and could not allow such applications? I know several large Fortran codes that still had many applications despite horrible coding.*
  **Authors**: It is not related to Fortran or coding style, and we also do not suggest this is an issue in TM5 or the former CarbonTracker code. It is more versatile because it is not integrated in the code of a specific transport model (TM5) and the transport model can therefore easily be swapped out for a different one, even Lagrangian/regional transport models are an option with CTDAS as stated in Section 2.3.

- *p. 2, l. 11: computational time has nothing to do with the number of code lines.*
  **Authors**: True. We did not mean to claim this and have removed the part on computational time, which is not relevant in this paragraph.

- *p. 3, l. 15 and 17: "error" is missing before "covariance" at both places.*
  **Authors**: Added.

- *p. 3, l. 19: if the system is robust for Europe and Northern America only, why are results for other parts of the globe shown (e.g. Fig. 8)?*
  **Authors**: We did not write that it is 'only' robust for Europe and North America, but meant to say that these regions were the focus area of 2 previous publications. We have rewritten the sentence.

- *p. 4, l. 3: from a quick check of the ECMWF web site, the situation of OOPS at ECMWF seems to be less advanced than what is suggested here (http://www.ecmwf.int/sites/default/files/elibrary/2017/17179-strategy-data-assimilation.pdf).*
  **Authors**: The sentence has been updated to say that the OOPS framework is still under development.

- *p. 10, l. 10: "Olson ecoregion" is not a standard expression.*
  **Authors**: We have included the citation repeated from the previous paragraph.

- *p. 10, l. 34: I could not find the 2.12 factor in Prather et al. (2012). The reference may be wrong.*
  **Authors**: Prather et al. (2012) is generally used as the main reference for this conversion (e.g. Le Quéré et al., 2016 and Ciais et al., 2013 (IPCC)), because it includes the value of 0.1765 Teramoles per ppb of dry air, which is required to calculate the conversion of ppm to PgC. This is explained step by step in Joos et al. (2013), and for clarity this reference has been added.

- *p. 11, l. 1: "well" should be quantified.*
  **Authors**: Agreed. The remaining difference has been quantified and added to the sentence.
  *What is the scientific meaning of the error bars in the figure if they are not properly computed (also for Fig. 7)?*
  **Authors**: The error bars are calculated properly. However, given that they do not include temporal covariances from week-to-week in our system, they are larger than we know they would be if we had this covariance available. A better estimate of the 'true' uncertainty of our flux estimates –the aim of our efforts– is by looking at a range of realizations, as discussed in Section 3.4.

- *p. 11, l. 1-2: the authors forget the role of transport model errors and the assumptions behind the NOAA estimates.*
  **Authors**: Indeed this is only part of the explanation. We have added 'e.g.' to indicate that, and added some additional explanative factors.

- *p. 12, l.22-23: how can this feature be an advantage? I would think otherwise.*
  **Authors**: The short window and absence of temporal correlation prevents the formation and persistence of dipoles in poorly observed regions, and makes our system less susceptible to large-scale transport model biases that can drive correlations between northern hemispheric and tropical carbon uptake (Stephens et al., 2007) . The sentence has been slightly reworded.

- *p. 12, l. 23: comparing a range with a standard deviation is not trivial. How is this done? Is the range assumed to represent 4 sigmas, 6 sigmas, …?*
  **Authors**: The range is not compared to a standard deviation. but indicates the minimum–maximum interval of the flux estimate.

- *p. 12, l. 24: this statement is valid only under the requirement that the realizations are of the same quality level.*
  **Authors**: No, since realizations of poorer quality would lead to poorer flux estimates. The spread in the flux estimates from different realizations is a measure for how well we know the fluxes (see also reply above to the general comments).

- *p. 13, l. 29: this result does not seem consistent with the plan to further shorten the assimilation window (p. 15, l.9).*
  **Authors**: We agree this statement was unclear. The window is the combination of the cycle length (currently 1 week) and the lag of the system (currently 5 times 1 week). We have rewritten the sentence to 'different' instead of 'shorter'.

- *p. 13, l. 31: the statement is intriguing and I could not find its origin in the Babenhauserheide et al paper. In its section 5.1.1, the latter paper discusses rejection and error assignment issues rather than optimization methods per se.*
  **Authors**: We meant the 'TM5-4DVar' setup specifically instead of '4DVar' in general and have updated the sentence. The observational coverage is discussed in the last paragraph of Section 5.1.1 and in the conclusions Section 6 of Babenhauserheide et al. (2015). The larger correlation (rewritten from 'covariance') between regions is discussed in the last paragraph of Section 2.2 and the compensation fluxes described in the first paragraph of Section 5.1.1 show one of the artifacts it creates (Babenhauserheide et al., 2015).

- *p. 13, l. 33: the statement seems to be too trivial for a "demonstration". "Illustrate" would be better, or am I missing something?*
  **Authors**: We agree that the use of "demonstrated" is overdone and the sentence has been reworded. The publication cited (van der Laan-Luijkx et al. 2015) includes inversion results where tropical observations (specifically in the Amazon) have been either included or excluded and shows that excluding these observations leads to a poorer match to observations, even compared to a simulation using prior fluxes, suggesting that the tropical fluxes act as the residual to close the carbon budget.

- *p. 14, l. 30: why did the use of the new Python shell need to be demonstrated in the first place?*
  **Authors**: As shown in the manuscript, and discussed above in reply to the general comments, the new Python shell allows for flexible setup and a wider range of applications compared to the former version of CarbonTracker, which was embedded in the TM5 transport model's code. The new shell CTDAS is being used in a wide range of applications already as shown in Section 4: for example for other gases $CH_4$ (Tsuruta et al. 2017), for regional application with a different transport model (He et al., in prep., Liu et al., in prep.), or for multi-tracer applications including carbon isotopes (van der Velde et al., in prep.). With the new shell these applications were more easily implemented, and more importantly not possible with the former version in case of switching to a new transport model (He et al., in prep., Liu et al., in prep.). The main reason to demonstrate the shell is therefore to document the shell so that it can be referred to when used in other applications, avoiding multiple descriptions of the shell in different upcoming papers. A model description paper in GMD seems the most logical way to accomplish this.

**References**

Babenhauserheide, A., et al.: Comparing the CarbonTracker and TM5-4DVar data assimilation systems for $CO_2$ surface flux inversions, Atmos. Chem. Phys., 15(17), 9747–9763, doi:10.5194/acp-15-9747-2015, 2015.

Chevallier, F., et al.: $CO_2$ Surface Fluxes at Grid Point Scale Estimated From a Global 21 Year Reanalysis of Atmospheric Measurements, J. Geophys. Res.-Atmos. 115, doi:10.1029/2010JD013887, 2010.

Chatterjee, A, and A M Michalak: Technical Note: Comparison of Ensemble Kalman Filter and Variational Approaches for $CO_2$ Data Assimilation. Atmos. Chem. Phys. 13 (23), 11643–60. doi:10.5194/acp-13-11643-2013, 2013.

Ciais, P., et al.: Carbon and Other Biogeochemical Cycles. In: *Climate Change 2013: The Physical Science Basis. Contribution of Working Group I to the Fifth Assessment Report of the Intergovernmental Panel on Climate Change* [Stocker, T.F., D. Qin, G.-K. Plattner, M. Tignor, S.K. Allen, J. Boschung, A. Nauels, Y. Xia, V. Bex and P.M. Midgley (eds.)]. Cambridge University Press, Cambridge, United Kingdom and New York, NY, USA, 2013.

He, W. et al.: CTDAS-Lagrange v1.0: A high-resolution data assimilation system for regional carbon dioxide observations, in preparation, 2017.

Joos, F., et al.: Carbon dioxide and climate impulse response functions for the computation of greenhouse gas metrics: A multi-model analysis, Atmos. Chem. Phys., 13(5), 2793–2825, doi:10.5194/acp-13-2793-2013, 2013.

Kang, J., et al.: Estimation of Surface Carbon Fluxes with an Advanced Data Assimilation Methodology, Journal of Geophysical Research-Oceans and Atmospheres 117 (D24), D24101. doi:10.1029/2012JD018259, 2012.

Le Quéré, C., et al.: Global Carbon Budget 2016, Earth Syst. Sci. Data, 8(2), 605–649, doi:10.5194/essd-8-605-2016, 2016.

Liu, Y., et al.: CarbonTracker Switzerland: Quantifying the net terrestrial biospheric carbon fluxes and uncertainties in central Europe and Switzerland for 2013, in preparation, 2017.

Peters, W., et al.: An ensemble data assimilation system to estimate $CO_2$ surface fluxes from atmospheric trace gas observations, J. Geophys. Res., 110(24), 1–18, doi:10.1029/2005JD006157, 2005.

Peters, W.,: An atmospheric perspective on North American carbon dioxide exchange: CarbonTracker, P Natl Acad Sci USA, 104(48), 18925–18930, doi:10.1073/pnas.0708986104, 2007.

Peters, W.,: Seven years of recent European net terrestrial carbon dioxide exchange constrained by atmospheric observations, Glob. Change Biol., 16(4), 1317–1337, doi:10.1111/j.1365-2486.2009.02078.x, 2010.

Peylin, P., et al.: Global atmospheric carbon budget: Results from an ensemble of atmospheric $CO_2$ inversions, Biogeosciences, 10(10), 6699–6720, 2013.

Stephens, B. B., et al.: Weak northern and strong tropical land carbon uptake from vertical profiles of atmospheric $CO_2$, Science, 316(5832), 1732–1735, 2007.

Tsuruta, A., et al.: Global methane emission estimates for 2000–2012 from CarbonTracker Europe-$CH_4$ v1.0, Geosci. Model Dev., 10(3), 1261–1289, doi:10.5194/gmd-10-1261-2017, 2017.

van der Laan-Luijkx, I. T., et al.: Response of the Amazon carbon balance to the 2010 drought derived with CarbonTracker South America, Global Biogeochem. Cy., 29(7), 1092–1108, doi:10.1002/2014GB005082, 2015.

van der Velde, I. R., et al.: Continent-wide increase of water-use efficiency in vegetation during large droughts of the recent decade, in preparation, 2017.